# A Prospective Cohort Study on Quality of Life among the Pediatric Population after Surgery for Recurrent Patellar Dislocation

**DOI:** 10.3390/children8100830

**Published:** 2021-09-22

**Authors:** Alexandru Herdea, Vlad Pencea, Claudiu N. Lungu, Adham Charkaoui, Alexandru Ulici

**Affiliations:** 1Pediatric Orthopedics “Grigore Alexandrescu” Children’s Emergency Hospital, 011743 Bucharest, Romania; alexherdea@yahoo.com (A.H.); vladpencea@gmail.com (V.P.); alexandru.ulici@umfcd.ro (A.U.); 2Department of Surgery, Country Emergency Hospital Braila, 810249 Brăila, Romania; lunguclaudiu5555@gmail.com; 3Morphological and Functional Sciences Department, Faculty of Medicine and Pharmacy, University of Galați, 800008 Galati, Romania

**Keywords:** recurrent patellar dislocation, knee injury, medial patellofemoral ligament reconstruction

## Abstract

Patellofemoral instability is a frequent cause of knee pathology affecting quality of life among the pediatric population. Here, we present a prospective cohort study which included patients who had undergone surgical management using the lateral release and medial imbrication approach (LRMI) or medial patellofemoral ligament reconstruction (MPFL-R). The object of this study was to assess the quality of life among children that have undergone surgical treatment for patellar dislocation. Quality of life was assessed before and after surgery using the Pediatric International Knee Documentation Committee form (Pedi-IKDC), a questionnaire that aims to quantify knee functionality. Postoperative scarring was evaluated using The Stony Brook Scar Evaluation Scale. One hundred and eight patients were selected and grouped according to the type of procedure. Before surgery, the two groups had similar mean Pedi-IKDC scores (41,4 MPFL-R vs. 39,4 LRMI *p* = 0.314). Improvements were observed in the postoperative scores. The MPFL-R technique showed promising outcomes. When comparing the two surgical groups, there was a significant difference in favor of MPFL-R group (MPFL-R 77.71 points vs. LRMI 59.74 points, *p* < 0.0001–95% CI (11.22–24.72)). Using the Stony Brook Scar Evaluation Scale, a significant difference in scar quality in favor of MPFL-R was observed (4,5 MPFL-R vs. 2,77 LRMI *p* = 0.002). In conclusion, this study provides objective evidence-based outcome assessments that support the medial patellofemoral ligament reconstruction technique as the gold standard for patellofemoral instability.

## 1. Introduction

Patellofemoral instability is a frequent cause of knee injury that occurs in the pediatric population [1,2]. The incidence rate is 29–43 per 100,000 individuals. The incidence of chronic instability is exceptionally high among girls between 10 and 17 [3]. The dynamics of the patellofemoral joint depends on both bony and soft tissue structures [4]. Therefore, developmental anomalies, traumatic disruption of static restraints, and weak dynamic stabilizers can lead to symptomatic instability [5,6]. Osteochondral fractures are an infrequent accompanying injury which can be successfully managed with the Steadman technique [7]. Some patients may benefit from platelet-rich plasma (PRP) injections in order to reduce the pain caused by injury to other structures of the knee such as the meniscus [8].

Clinical diagnosis is mainly based on the medical history of patellar dislocation and the extent of the hemarthrosis that must be evacuated to reduce pain [9]. In order to correctly assess a patellofemoral instability, clinical examination, conventional X-rays, and C.T. or MRI are needed [10]. However, in most severe cases, computed tomography followed by 3D reconstruction and 3D printing can help the orthopedic surgeon to plan the safest and the most effective surgical approach [11].

Conservative treatment usually consists of cast or splint immobilization, resulting in longer rehabilitation periods as well as a recurrence rate of up to 44% [6,8]. Surgical treatment is the next recommended step if conservative management fails to improve the symptoms significantly. Surgery is recommended in the case of recurrent dislocation [12].

Two popular surgical treatments are lateral release with medial imbrication (LRMI) and medial patellofemoral ligament reconstruction (MPFL-R). Lateral release is sometimes also performed along with MPFL-R to reduce the pull of the lateral retinaculum in order to decrease the stress placed on the medial retinaculum, and is an especially useful technique in pediatric patients [13]. However, there is conflicting information in the literature regarding LRMI, with several recent studies demonstrating good outcomes following application of the technique. In contrast, other studies have shown a high failure rate and a high occurrence of complications [14,15]. MPFL-R aims to restore the normal anatomy of the knee joint with either a autograft or a synthetic graft. Because the MPFL is the main restraint to lateral dislocation in the first 30° of flexion, proper reconstruction will prevent the recurrence of dislocation and prevent undue stress on the knee caused by an abnormal anatomy [16].

The International Knee Documentation Committee Pediatric (IKDC-Pedi) questionnaire has been shown to be relevant is assessments of patient QoL in a variety of knee injuries, including patellar dislocation [17].

## 2. Materials and Methods

The purpose of the study was to assess the QoL of patients that suffered from episodic patellar dislocation and were treated using LRMI or MPFL-R with a double bundle synthetic graft. The average patient age at diagnosis of patellar dislocation was 13.3 years ± 2 years; see Figure 1 Most patients (96%) had at least two more luxation episodes between diagnosis and surgery.

The study was carried out on 108 pediatric patients (aged 10–18) that had undergone either LRMI or MPFL-R between 2013–2018. The diagnosis was established based on clinical findings, radiologic exams, and magnetic resonance imaging scans, using the following inclusion criteria: history of multiple locked dislocations or locked dislocation present at admission, presence of hemarthrosis, positive apprehension test, painful medial parapatellar structures, and femoral epicondyle, as well as a minimum follow-up of 24 months. Exclusion criteria were: avulsion fracture or femoral condyle osteochondral fracture, lack of preoperative and postoperative knee radiographs, or lack of informed consent. Knee radiographs performed in the anteroposterior and lateral view were used to identify complications. Magnetic resonance imaging (MRI) was used to evaluate soft tissue lesions and to determine the treatment plan by examining the growth plate and assessing whether additional procedures were needed, such as trochleoplasty or patellar tendon realignment. Both surgeries have similar indications, namely, recurrent patellar dislocation with severe trochlear dysplasia. Postop complications that would affect patient outcomes include recurrent dislocation or pain due to the altered knee anatomy; however, the latter occurs mainly in LRMI. LRMI also presents a risk of overly reducing lateral forces on the patella, thus inducing medial dislocation, worsening the patient’s QoL and requiring further corrective surgery [18].

Patients were randomly assigned to a surgical group in the following manner: those diagnosed on an even date were assigned to LRMI while those diagnosed on an uneven date were assigned to MPFL-R. Following the randomization, 80 patients were assigned to the LRMI group and 28 to the MPFL-R group. The mean age at surgery was 14.2 years ± 2 years in the LRMI group and 14.5 years ± 2 years in the MPFL-R group. There were no statistically significant differences in age (*p* = 0.091) or sex (*p* = 0.07); see Figure 1.

The postoperative rehabilitation protocol consisted of 1 week of avoiding weight-bearing movements on the operated knee, with subsequent physiotherapy with the purpose of increasing knee stability and proprioception.

Quality of life was evaluated before and after surgery using the Pediatric International Knee Documentation Committee (IKDC-Pedi) form. The postoperative evaluation of the quality of life was conducted after 24 months of follow-up. The average interval from surgery to follow-up was 30 months (25–50 months). Postoperative scarring was also assessed using The Stony Brook Scar Evaluation Scale (SBSES). Patients filled out the questionnaires under parental guidance in the presence of the attending physician.

For statistical analysis, we assumed a null hypothesis of equal efficacy of MPFL-R and LRMI. We set the significance level at 5% (0.05). We modeled the frequency by running a Shapiro-Wilk Test. As the data went through normal distribution, the independent Student-T test was used to compare IKDC-Pedi scores between patients who had undergone LRMI surgery and those who experienced MPFL-R. The response to the athletic ability-related question on the Pedi-IKDC form could not be used to express a normal distribution, so a Mann-Whitney U test was run to check for statistical significance. As the scar evaluation data was not equally distributed, a Mann-Whitney U test was also needed. Standard deviation (S.D.) was calculated, and a confidence interval (CI) of 95% was used.

The acquired and statistically analyzed data comprised the following variables: age, sex, type of surgery, date of surgery, athletic level, preoperative IKDC-Pedi score, postoperative IKDC-Pedi score, postoperative The Stony Brook Scar Evaluation Scale.

## 3. Results

A total number of 130 patients were operated on for episodic patellar dislocation in the selected time interval. Five of them were excluded from the study due to a lack of adequate postoperative radiographs. Twelve more were excluded because they underwent other, subsequent surgical techniques. Five patients did not consent to take part in the study. One hundred and eight recreational athletes fulfilled the inclusion and exclusion criteria, completed the questionnaires, and presented at follow-up (see flow chart below-Figure 2).

Preoperatively, the Pedi-IKDC scores were similar between the two surgical groups (MPFL-R 41.4 points vs. LRMI 39.4 points), and the difference was not statistically significant (*p* = 0.314). We found significant improvement following both surgical approaches, with the MPFL-R group scoring better than LRMI in postoperative IKDC-Pedi forms compared to preoperative assessment (MPFL-R + 36.36 points-95% CI (27.76–44.97) vs. LRMI +20 points-95% CI (15.11–25.53), *p* < 0.0001). A statistically significant difference in the postoperative IKDC-Pedi score between the two groups (MPFL-R 77.71 points vs. LRMI 59.74 points, *p* < 0.0001-95% CI (11.22–24.72)) was observed (see Figure 3).

There was also a statistically significant difference regarding the pain related questions of the Pedi-IKDC questionnaire, favoring MPFL-R (MPFL-R 15.8 points vs. LRMI 12.3 points *p* = 0.00175 95% CI (1.36–6.21).

## 4. Discussion

The patients from the MPFL-R group had significantly better IKDC-Pedi scores as well as significantly better scar quality. The different Pedi-IKDC scores were primarily tied to patient ability to improve or return to their previous activity level. The questions related to athletic ability showed the most significant differences in favor of MPFL-R. One possible explanation may be the faster mobilization postsurgery, which would protect against the muscular atrophy caused by immobilization [19].

IKDC-Pedi was chosen as the QoL measurement because it had better responsiveness than KOOS-Child. In addition, as a shorter questionnaire makes, it is more likely to be fully completed in a clinical setting [20].

The Stony Brook Scar Evaluation Scale (SBSES) was selected for the same reasons: the short time required for its completion and its good clinical relevancy [21].

This is because MPFL-R restores the normal anatomy of the knee, thus facilitating regular joint reaction forces [18]. However, in LRMI, the increased joint forces could cause unpleasant sensations like pressure or pain in the knee joint [22].

The postoperative score for the MPFL-R group correlated with data from other studies found in literature, indicating good surgical technique and rehabilitation programs [23]. The LRMI group had fewer reported redislocations than most studies using a similar surgical technique, yet the IKDC-Pedi score was lower than expected [24,25].

The cosmetic differences between the two procedures are also undeniable. MPFL-R is far less invasive and results in a significantly better-looking postsurgical scar. The SBSES does not consider scar length, and it should be mentioned that the MPFL-R group has two short scars while the LRMI has one long scar. As observed in our study, this is a cause of distress for patients even if the scar itself has healed without abnormal pigmentation, elevation, or depression.

One significant factor in this patient group is tibial and femoral physis [26]. While the surgical technique used for MPFL-R in this study does not usually affect the growth plate, there is still a slight risk. In contrast, the LRMI procedure only involves the soft tissues surrounding the knee, eliminating any risk of damage to the growth plate.

Lateral release on its own has yielded unsatisfactory outcomes in the history of pa-llar dislocation treatment [27], and release of a normal lateral retinaculum may increase lateral patellar translation and cause even more instability due to the role of the lateral retinaculum in resisting lateral patellar translation [28]. In one study which compared MPFL-R without lateral release and MPFL-R with a lateral release, the groups had similar outcomes, thus showing that lateral release is not mandatory [29].

The follow-up period was clinically relevant because most redislocations (70%) occur within 24 months postoperatively [30]. However, it is insufficient to determine whether differences in joint anatomy that result after surgery remotely affect the incidence of osteoarthritis. This would require a lengthy, hard-to-manage longitudinal study. However, when considering the fact that patellar instability very often leads to unfavorable outcomes in adults, either surgery is desirable compared to nonsurgical treatment [31].

The strengths in our study were the homogeneity of the surgical techniques that were identical for all patients and the homogeneity of the study groups concerning risk factors for patellar dislocation. The limitations were the unequal treatment groups as well as differences in follow-up period.

## 5. Conclusions

MPFL-R increased patient quality of life more than LRMI. MPFL-R interventions are minimally invasive, reduce postoperative recovery time and increase quality of life. This study provides further evidence for the recommendation of MPFL-R as the gold standard for patellofemoral instability. However, further studies are needed to observe the long-term stability and side effects of MPFL-R.

## Figures and Tables

**Figure 1 children-08-00830-f001:**
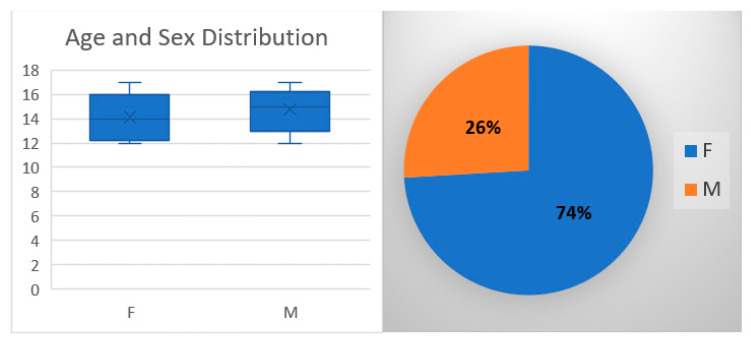
Age and sex distribution.

**Figure 2 children-08-00830-f002:**
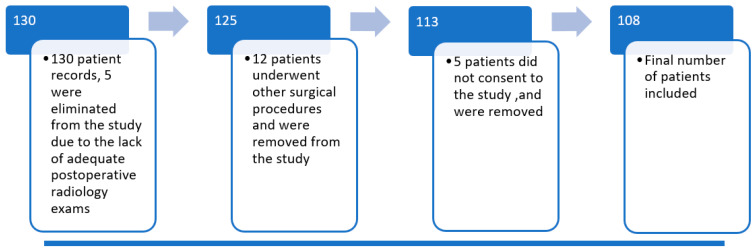
Flow diagram of the patients included in the study. After inclusion and exclusion criteria were applied, 108 patients remained in the study.

**Figure 3 children-08-00830-f003:**
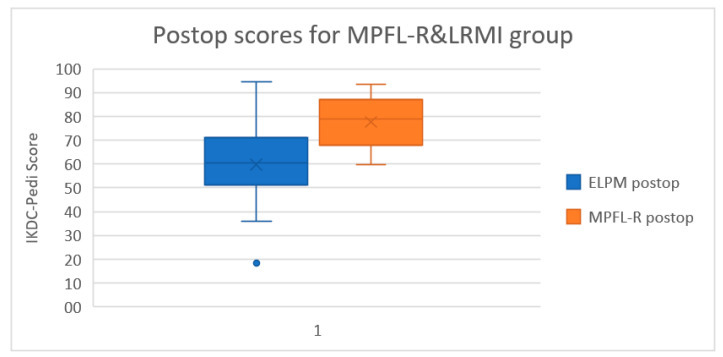
Postoperative scores for MPFL-R and LRMI group. The *p* values are represented in the label for each group.

## Data Availability

The datasets used and analyzed during the current study are available from the corresponding author on reasonable request.

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
