# Peer review of "A Prospective Cohort Study on Quality of Life among the Pediatric Population after Surgery for Recurrent Patellar Dislocation"

_children, 2021, doi:10.3390/children8100830_

Round 1

Reviewer 1 Report

This is an interesting clinical study comparing the outcome and patient satisfaction following 2 techniques of patellar instability treatment.

Abstract

In its current form it is not acceptable.

Merge lines 2 and 3. Present the results of your study in a more appropriate way, i.e. report the results with numbers.

Describe, more accurately, the two study groups and the evaluation tools.

Introduction

The introduction is extremely extended and general and should be reduced to 4-5 paragraphs.

The first paragraph should briefly describe the incidence and pathoanatomy of patellar dislocation, the second the accompanying injuries, the third the indications for surgical reconstruction, the fourth the available surgical techniques and the final paragraph should focus on the study questions.

Paragraph 1, Line 1. Please improve. I think that paragraphs 1, 2 and 3 are not necessary.

Paragraphs 5,6,7, 8 are not necessary.

Please rewrite and restructure the introduction to make it presentable.

Materials and methods

The first sentence is incomplete,

Please be more specific regarding the number of patients, the selection of the surgical technique, its brief description, the post regime, and the evaluation methods. Proved demographic data and compare the study groups. Describe the indications for each technique and potential postoperative complications which may have affected the result.

Line 4, pediatric patients (age 0-18). Please be more specific.

Statistical analysis is well documented.

Results

You are reporting that a randomization process was performed. Please provide more accurate information.

There were no statistically significant differences in age, sex, or athletic level. Please be more specific and provide numbers.

Discussion

The first paragraph should describe the main findings of the study.

Paragraph 4. In our study, we noticed better outcomes related to pain assessment among the 190 MPFL-R group. This statement belongs to the results section and should be more clearly presented.

The discussion should be more concise and structured to focus only on the main results of the study.

Figure 1. It is not clear enough. Please include the final number of patients, 108, in the flow diagram.

Figure 2. I think it would be better to combine both charts into one for comparison purposes.

Figure 3. correct preporatory. On the graph, only postoperative scores are shown.

Author Response

Comment 1: abstract Merge lines 2 and 3. Present the results of your study more appropriately, i.e., report the results with numbers.

Response 1: Lines 2 and 3 were merged as suggested. Numbers were included.

Comment 2: Describe, more accurately, the two study groups and the evaluation tools.

Response 2:  Lines 25 to 30 were removed and replaced with the following text: Before surgery, the two groups had similar mean Pedi-IKDC score(41,4 MPFL-R vs 39,4 LRMI p=0,314). Improvement in the postoperative score after surgery was observed. The Medial Patellofemoral Ligament Reconstruction technique showed promising outcomes. When comparing the two surgical groups there was a significant difference in favor of the MPFL-R group (MPFL-R 77.71 points vs. LRMI 59.74 points, p<0.0001 - 95% CI [11.22-24.72]). Using the Stony Brook Scar Evaluation Scale, a significant difference in scar quality in favor of MPFL-R was observed (4,5 MPFL-R vs. 2,77 LRMI p=0,002).

Comment 3: introduction The introduction is highly extended and general and should be reduced to 4-5 paragraphs.

Response3: The introduction structure was modified according to suggestions

Comment 4: The first paragraph should briefly describe the incidence and pathoanatomy of patellar dislocation, the second the accompanying injuries, the third the indications for surgical reconstruction, the fourth the available surgical techniques and the final paragraph should focus on the study questions.

Responbse 4 : Paragrepahs were structured accordingly.

Comment5: Paragraph 1, Line 1. Please improve. I think that paragraphs 1, 2 and 3 are not necessary.

Response5: corrections have been made as suggested.

Comment6: Paragraphs 5,6,7, 8 are not necessary.

Response6: paragraphs were deleted as suggested

Comment7: Please rewrite and restructure the introduction to make it presentable.

Respomnse 7: The introduction has beeb rerwitten

Comment8: Materials and methods The first sentence is incomplete, Please be more specific regarding the number of patients, the selection of the surgical technique, its brief description, the post regime, and the evaluation methods. Proved demographic data and compare the study groups. Describe the indications for each technique and potential postoperative complications which may have affected the result.

Response 8:  Text was modified accordingly

Comment9: Line 4, pediatric patients (age 0-18). Please be more specific.

Comment9: text was modified accordingly

Comment10: Results You are reporting that a randomization process was performed. Please provide more accurate information.

 Response10: text was modified accordingly

Comment11: results There were no statistically significant differences in age, sex, or athletic level. Please be more specific and provide numbers.

Response11: text was modified accordingly

Comment 12:  discussions The first paragraph should describe the main findings of the study.

Response12: paragraph was modified accordingly.

Comment 13: discussion Paragraph 4. In our study, we noticed better outcomes related to pain assessment among the 190 MPFL-R group. This statement belongs to the results section and should be more clearly presented.

Response13: the statement was removed

Comment14: discussions The discussion should be more concise and structured to focus only on the main results of the study.

Comment 15: discussions Figure 1. It is not clear enough. Please include the final number of patients, 108, in the flow diagram.

Response15: diagram was modified.

Comment16: discossionsFigure 2. I think it would be better to combine both charts into one for comparison purposes.

Response: the figure was modified

Comment17 discussions: Figure 3. correct preparatory. On the graph, only postoperative scores are shown.

Response17: corrected.

Reviewer 2 Report

Overview: The topic of the article is interesting. However, I’m not sure if the authors wanted to assess the quality of life (I haven’t seen any questionnaire for quality of life assessment used in this study) or they are just comparing clinical results of two surgical methods measured in Pedi-IKDC. It should be written clearly in the text. It will be a prospective, single-center, study. The study wasn’t randomized and the exact follow-up is not known. The introduction, methods, and discussion should be improved.

Nevertheless, I have some suggestions to improve the paper:

  1. The English language should be corrected by a native speaker or professional translator because there are many mistakes.
  2. Introduction: This section should be shortened and improved. Authors should write the aims of the study. Pedi-IKDC is a questionnaire for assessing the clinical results of knee surgery in the young population, not for assessing the quality of life.
  3. Methods and results: The number of patients in the LRMP group was 80 and 28 in the MPFL-R group. I think that the difference between the groups is too high. Figure 2 is used inappropriately. Please add circle diagrams or remove them. The range of follow-up should be added.
  4. I suggest adding the CONSORT flow diagram of the trial.
  5. The discussion section should be improved. Please compare your results with other works on a similar topic.
  6. The conclusion section should resume the results. The information contained in this section doesn’t follow the results of this study.

Yours sincerely

Author Response

Comment1: The English language should be corrected by a native speaker or professional translator because there are many mistakes.

Response1: the English language was corrected

Comment2: Introduction: This section should be shortened and improved. Authors should write the aims of the study. Pedi-IKDC is a questionnaire for assessing the clinical results of knee surgery in the young population, not for assessing the quality of life.

Response 2: Although Pedi-IKDC is a questionnaire for assessing the clinical results of knee surgery in the young population is was found to be appropriate to be used in assessing the quality of life

  1. Reproducibility and responsiveness of a Danish Pedi-IKDC subjective knee form for children with knee disorders J. S. Jacobsen,P. Knudsen,C. Fynbo,N. Rolving,S. Warming, https://doi.org/10.1111/sms.12589

2.The Pediatric International Knee Documentation Committee (Pedi-IKDC) Subjective Knee Evaluation Form: Normative Data Adam Y. Nasreddine, MA,1 Susan Elizabeth Nelson, MD,MPH,2 Patricia Connell, MPH,3 Leslie Kalish, DSc,3 and Mininder S. Kocher, MD, MPH3

Comment3: Methods and results: The number of patients in the LRMP group was 80 and 28 in the MPFL-R group. I think that the difference between the groups is too high. Figure 2 is used inappropriately. Please add circle diagrams or remove them. The range of follow-up should be added.

Response3: the figure was corrected.

Comment4:I suggest adding the CONSORT flow diagram of the trial.

Response 4: This study was conducted accordingly  to STROBE  

Comment5: The discussion section should be improved. Please compare your results with other works on a similar topic.

Response5: The discussion were performed in accordance with other works. References 19-32.

Comment6: The conclusion section should resume the results. The information contained in this section doesn’t follow the results of this study.

Response: Conclusions resume the results.

Reviewer 3 Report

Dear Authors

I have reviewed your paper with great interest.

I will accept your paper after a minimal revision.

My revision is:

Title: Very Good

Abstract: Very Good

Introduction and AIM: The problem and the aim are well descripting.

Marterials, Patients and methods and statistics: All good.

Results: Focus on and well described.

Discussion and Thread: effectiveness Focus ON.

Patellar instability can prodiuce fast arthrosis or permanent instability with bad outcomes in adults, please cite and discuss these papers:

Bisaccia M., Caraffa A., Meccariello L., Ripani U., Bisaccia O., Gomez-Garrido D., Carrado-Gomez M., Pace V., Rollo G., Giaracuni M., Rinonapoli G. Displaced patella fractures: percutaneous cerclage wiring and second arthroscopic look. Clin Cases Miner Bone Metab. 2019;16(1):48-52.

Pellegrino M, Trinchese E, Bisaccia M, Rinonapoli G, Meccariello L, Falzarano G, Medici A, Piscitelli L, Ferrara P, Caraffa A.  Long term outcome of grade III and IV chondral injuries of the knee treat with Steadman microfracture technique. Clin Case in Mineral and Bone Metabolism 2016; 13(3):237-240

References: Well chosen but to improve

Figures and Table: Very Good.

Author Response

Comment1: Patellar instability can prodiuce fast arthrosis or permanent instability with bad outcomes in adults, please cite and discuss these papers:

Bisaccia M., Caraffa A., Meccariello L., Ripani U., Bisaccia O., Gomez-Garrido D., Carrado-Gomez M., Pace V., Rollo G., Giaracuni M., Rinonapoli G. Displaced patella fractures: percutaneous cerclage wiring and second arthroscopic look. Clin Cases Miner Bone Metab. 2019;16(1):48-52.

Pellegrino M, Trinchese E, Bisaccia M, Rinonapoli G, Meccariello L, Falzarano G, Medici A, Piscitelli L, Ferrara P, Caraffa A.  Long term outcome of grade III and IV chondral injuries of the knee treat with Steadman microfracture technique. Clin Case in Mineral and Bone Metabolism 2016; 13(3):237-240

Response 1 references 7 and 32 were added.

Comment2: references Well-chosen but to improve

Response 2: references were improuved

Round 2

Reviewer 1 Report

the corrections are acceptable.

Author Response

Thank you for reviewing the manuscript. 

Reviewer 2 Report

Dear Authors,

You did some corrections that I’ve suggested. However, some important issues haven’t been done.

I think that the article should be improved:

  1. The limitations of the study should be included in the text (e.g difference in the number of patients included between the groups, differences in follow-up period)
  2. The conclusions section should be improved. The main result of this study was that the quality of life was improved after both surgical approaches.

Yours sincerely

Author Response

Thank you for reviewing our article.

We have made the requested modifications and responded to each one.

Comment 1: The limitations of the study should be included in the text (e.g., the difference in the number of patients included between the groups, differences in follow-up period)

Response1:  Limitations of the study were stated in the manuscript. The following text was added at the discussion section: ,, The limitations of this study are the unequal treatment groups as well as differences in follow-up period'' Line  307-308

Comment 2: The conclusions section should be improved. The main result of this study was that the quality of life was improved after both surgical approaches.

Response2: Both surgical procedures improve the quality of life.  The following text was added to the conclusion section: ,, MPFL-R increased the patients' quality of life more than LRMI''. Line 311